# REST/NRSF Silencing Modifies Neuronal Gene Expression in siRNA-Treated HeLa Cells: A Preliminary Exploration in the Search for Neuronal Biomarkers of Cervical Cancer

**DOI:** 10.3390/medicina59030537

**Published:** 2023-03-09

**Authors:** Karen Cortés-Sarabia, Luz Del Carmen Alarcón-Romero, Miguel Ángel Mendoza-Catalán, Juan Carlos Carpio-Pedroza, Eduardo Castañeda-Saucedo, Carlos Ortuño-Pineda

**Affiliations:** 1Facultad de Ciencias Químico-Biológicas, Universidad Autónoma de Guerrero, Av. Lázaro Cárdenas s/n, Chilpancingo 39090, Mexico; 2Departamento de Parasitología, Instituto de Diagnóstico y Referencia Epidemiológicos “Dr. Manuel Martínez Báez”, Francisco de P. Miranda 177, Mexico City 01480, Mexico

**Keywords:** neural phenotype, REST targets, transcriptional regulation, HeLa cells, cervical intraepithelial neoplasia

## Abstract

*Background and Objectives:* REST (RE1-silencing transcription factor) diminution is associated with transcriptional relaxation, neuropeptide overexpression, and phenotype redefinition in neuroendocrine cancers, but this effect has barely been studied in cervical cancer (CC). We previously reported reduced expressions of REST in samples with premalignant lesions and CC; however, the transcriptional consequences for neural genes associated with reduced REST expression in CC are unknown. Therefore, the objective of this work was to evaluate the expression of neuronal genes in cancerous cells with reduced expression levels of REST. *Materials and Methods*: Here, we monitored levels of REST by immunostaining along the premalignant lesions and in invasive cervical squamous cell carcinoma (SCC) and endocervical adenocarcinoma (ADC) in tissue samples from female patients from southern Mexico and the derivative cell lines SiHa and HeLa, respectively. Next, we selected REST target genes in silico and explored the effect of REST silencing by RT-PCR in siRNA-treated HeLa cells. *Results*: The results show a REST diminution in premalignant lesions, SCC, ADC, and cancerous cell lines. Further REST silencing in HeLa cells altered the expression of genes containing the RE1 (Restrictive Element 1) sequence, including CgA (chromogranin A), CHRNβ2 (cholinergic receptor nicotinic β 2 subunit), BDNF (brain-derived neurotrophic factor), CRF (corticotropin-releasing factor), and RASSF1A (Ras association domain family 1). *Conclusions*: This work provides preliminary evidence of the role of REST loss in the transcriptional regulation of its target genes in HeLa cells, which could have positive implications for the search for new biomarkers of cervical cancer.

## 1. Introduction

In the epithelial tissues of mammals, REST (also known as neuron-restrictive silencer factor, NRSF) binds to targets of the RE1 (Restrictive element 1) sequence and represses neural gene transcription by orchestrating chromatin inactivation. In contrast, the degradation of REST promotes neurogenesis via transcriptional relaxation during embryogenesis and neuronal phenotype maintenance in adults [1]. In different neural contexts, REST diminution involves feedback circuits, including microRNAs, splicing isoforms, chromatin remodelers, proteasomal degradation [2,3,4], cytoplasmic sequestration [5], and inactive REST4 oligomers, or REST4/REST hetero-oligomers [6].

The drastic impact of nuclear REST diminution on the alteration of genes containing the RE1 sequence is evidenced by the nearly 2000 genes that are potential targets for REST in mice, rats, and humans and supported by experiments where transfection with the antagonist REST4 allows the transition of cardiomyocytes to physiologically active neurons [7,8]. Physiologically, in neural embryonic stem cells (ESCs), the absence of REST allows the expression of L1CAM, SYP, SYN1, BDNF, GRIN2 A, and GPR123, which are involved in neural differentiation, axonal growth, vesicular transport, and ionic conductance [9]. Pathologically, in small cell lung [10], breast [11], and prostate cancer [12], abnormal diminution of REST is associated with neuroendocrine phenotype acquisition. These neuroendocrine phenotypes are characterized by the re-expression of neural markers such as neural cell adhesion molecule 1, arginine vasopressin, synaptophysin, and chromogranin A, some of which are under the transcriptional control of REST [13,14].

Cervical cancer is one of the most frequent malignant tumors in women. In 2020, 604,100 new cases and 341,800 deaths were reported worldwide [15]. As a result, the search for new biomarkers with diagnostic and prognostic value for cervical cancer has become an area of great interest. We previously reported that REST is reduced in human samples diagnosed with squamous intraepithelial lesions (SILs) and invasive cervical squamous cell carcinoma (SCC) of the cervix with high-risk human papillomavirus [16]. However, the impact of REST diminution on the transcription of its neural targets in CC is unknown. In this work, we propose the potential role of REST downregulation on the expression of neural genes, which can serve as target molecules in the diagnosis of cervical cancer.

## 2. Materials and Methods

### 2.1. Immunocytochemistry (ICC) and Immunohistochemistry (IHC)

For immunocytochemistry (ICC), Hela cells (ATCC^®^ CCL-2^TM^) and SiHa cells (ATCC^®^ HTB-35™) were obtained from the American Type Culture Collection (ATCC, Manassas, VA, USA) and grown on coverslips and fixed with cold methanol for 10 min. For immunohistochemistry, the tissue samples were collected from female patients from southern Mexico by a pathologist, specifically at the Instituto Estatal de Cancerología de Guerrero (IECan), and processed at the Universidad Autónoma de Guerrero (UAGro). The Ethics Committees of the UAGro and IECan (general agreement UAGRO-IECA, April 18, 2016) approved the study protocol, and all participants signed an informed consent in accordance with the Declaration of Helsinki. The histopathological diagnosis was performed with H and E staining and analyzed by an experienced pathologist based on the cervical intraepithelial neoplasia classification [17]. Paraffin tissue sections of 3 µm in thickness were cut from a paraffin block and deparaffinized with xylene. Before IHC, tissues were rehydrated in alcohol baths and water. In both cases, ICC and IHC, we used the streptavidin-biotin-peroxidase method. As a primary antibody, we used an anti-REST antibody (Santa Cruz Biotechnology Cat#Sc-25398, Dallas, TX, USA). Cells and tissues were subjected to antigen retrieval (ImmunoDNA Retriever Citrate) for 15 min at 120 °C. The incubation with the primary antibody diluted 1:100 was performed for 2 h, and then biotin and streptavidin peroxidase were added. The reaction was developed using the chromogen 3,3′-diaminobenzidine (DAB), counterstained with Mayer’s hematoxylin, and fixed with synthetic resin for future analysis. In the negative control, the primary antibody was omitted.

### 2.2. In Silico Analysis

Gene sequences analyzed in this study were downloaded from the nucleotide database of the NCBI. For RE1 quest in analyzed genes, we searched for previous reports performed by Kashiwagi et al. 2012 [18] and Schoenherr et al. 1996 [19] and performed pairwise sequence alignment in the EMBOSS needle server (https://www.ebi.ac.uk/Tools/psa/emboss_needle/nucleotide.html) (accessed on 1 March, 2019). Data for mRNA levels for REST, Chromogranin A, CREB, and Kif17 in cervical cancer were obtained from the Human Protein Atlas (https://www.proteinatlas.org/) (accesed on 1 April 2019).

### 2.3. HeLa Cell Culture and Transfection

HeLa cells were grown in DMEM: F-12 medium (ATCC^®^ 30-2006™) supplemented with 10% fetal bovine serum (ATCC^®^ 30-2020™) at 37 °C with 5% CO_2_. For transient transfection, cells were grown in 24-well plates at a density of 3 × 10^4^ cells/well. The small interfering RNA (siRNA) against REST (Life Technologies Cat#4392420, Carlsbad, CA, USA) and the negative control (Life Technologies Cat#4390843) were used at a concentration of 25 nM. Lipofectamine RNAiMAX (Invitrogen Cat#13778-150, Waltham, MA, USA) was used for transfection following the manufacturer’s instructions. After 72 h of incubation, cells were collected and used for Western blot and RT-PCR analysis.

### 2.4. Western Blot

Whole protein extracts from transfected HeLa cells were obtained by using a RIPA buffer (50 mM Tris, 150 mM NaCl, 0.1% SDS, 0.5% sodium deoxycholate, 1% NP-40, protease inhibitors, and PMSF). Samples were separated by SDS-PAGE and transferred to nitrocellulose membranes. Membranes were tested using antibodies against REST (dilution 1:1000; Santa Cruz Biotechnology Cat# sc-25398) or GAPDH (dilution 1:2000; Santa Cruz Biotechnology Cat# sc-365062) as described previously [3]. Anti-rabbit igG-B (dilution 1:5000; Santa Cruz Biotechnology Cat# sc-2040) and HRP-goat anti-mouse antibodies (dilution 1:5000; Invitrogen Cat# 62-6520) were used as secondary antibodies, respectively.

### 2.5. RT-PCR

Total RNA was extracted using TRIzol^TM^ according to the manufacturer’s instructions (Thermo Fisher Scientific Cat#15596026, Waltham, MA, USA). RNA was recovered in 50 µL of RNase-free water and stored at −20 °C until use. According to the manufacturer’s instructions, RT-PCR was performed using the oligonucleotides described in the results section and 1.5 µg of total RNA (Thermo Fisher Scientific, Cat#11146-016). The PCR products were visualized on 1.2% agarose gels stained with ethidium bromide.

## 3. Results

In addition to the evidence showing a consistent association between REST diminution and neural trait acquisition, we found a significant reduction in nuclear REST in cytology samples of premalignant lesions and cervical cancers from an extensive group of patients [16]. These findings suggest a potential effect of REST diminution on the transcriptome and re-expression of neural genes. However, the association between REST diminution and neural gene expression in these tumors has not been studied. To analyze this association, we first generated expression profiles of REST by immunostaining tissues corresponding to cervical intraepithelial neoplasia 1 (CIN 1, *n* = 9), cervical intraepithelial neoplasia 2/3 (CIN 2/3, *n* = 7), and cervical cancer (CC, *n* = 11), as well as HeLa and SiHa cell lines. The phenotypic characterization of tissues was performed using H and E staining by a pathologist and can be classified as follows: in CIN 1 cells with binucleation, perinuclear halo, and karyomegaly are observed, and in CIN 2/3 and CC cells, karyomegaly, dyskaryosis, hyperchromatic nuclei, and granular chromatin are observed. The results confirm the evident nuclear REST diminution in CIN 2/3 and cancer basal and parabasal cell layers. Interestingly, REST diminution was more significant in ADC than SCC (Figure 1A). Concordantly, the data from the Human Protein Atlas showed mRNA level dysregulation of the REST neural targets in tissues with aberrant levels of REST. These genes included CgA, CREB, Xbp1, RASSF1A, and CDH1 (Figure 1B). Next, we selected 11 REST target genes for expression analysis based on the presence of the RE1 sequence as reported by other groups and our in silico analysis. Such genes included CgA, CHRNβ2, BDNF, CHRM4, CRF, CREB, GluR1, Xbp1, Kif17, RASSF1A, and CDH1. The most conserved RE1 sequences were located equally in the promoter, introns, exons, and 5′UTR regions of the CgA, CHRNβ2, BDNF, CRF, CDH1, and RASSF1A genes, whereas the less conserved sites were in the GluR1, Xbp1, and Kif17 genes (Figure 2A). Finally, we treated HeLa cells for 72 h with siRNA to further reduce REST expression and analyze the transcription of the target genes (Figure 2B). The Western blot confirmed REST reduction after siRNA treatment (Figure 2C), and RT-PCR assays showed altered expression of neural genes. Specifically, there was a high expression of CHRNβ2, BDNF, and RASSF1A and a reduced expression of the CgA and CRF genes. We did not observe changes in the expression of genes with less conserved RE1 sequences (GluR1, CREB, RASSF1A, Xbp1, Kif, and CDH1 genes), suggesting that the transcriptional regulation of REST is selective (Figure 2C). Because we used the HeLa cell line as a model, the reliability of using such markers in diagnosis will depend on testing them extensively in a statistically appropriate number of patient samples.

## 4. Discussion

Some examples of cancers with reduced expression levels of REST include lung [10], breast [11], prostate [12], and skin cancers [20]. The nuclear REST diminution recently found in premalignant lesions and cervical cancers with high-risk human papillomaviruses appears to be a promising diagnosis tool [16]. However, the expression of REST at several stages of cervical carcinogenesis and the effect of REST diminution on the transcription of neuronal genes have not yet been studied. In the present study, we analyzed the expression profile of REST in CIN 1, CIN 2/3, SCC, and ADC, as well as the transcriptional effect of the reduced expression of REST on the expression of CgA, CHRNβ2, BDNF, CHRM4, CRF, CDH1, CREB, GluR1, Xbp1, Kif17, and RASSf1A genes in HeLa cells. Of note, all of these genes contain the RE1 sequence and, in some cases, are closely related to cancer development and progression [11,13,20].

CIN 2/3, SCC, and ADC samples, as well as cancerous cells, showed a diminution in their REST expression (Figure 1). Additionally, further REST diminution in HeLa cells by means of siRNA treatment altered the expression of neuronal target genes containing the RE1 sequence (Figure 2), suggesting a potential role for REST in the alteration of many neural genes in cancer with potential application in diagnosis. Although the analysis shown in Figure 1B mainly corresponds to data obtained from squamous cell carcinoma samples and the results shown in Figure 2C correspond to the HeLa cell line, which is derived from an adenocarcinoma, in both cases we observed alterations in the expression of some neuronal genes. Remarkably, an increase in the expression of the CHRNβ2 neuronal gene in HeLa cells was observed, which has been widely reported as a transcriptional target of REST and could be included in the panel of cervical cancer prognostic biomarkers. In the first instance, some of the neuronal genes with altered expression could be included in the panel of biomarkers whose protein levels could be observed by immunohistochemistry in a formalin-fixed, paraffin-embedded block of tissue using (commercially available) monoclonal antibodies against each protein. Additionally, an RT-PCR could be standardized to monitor the RNA levels of these genes in the cervical cytologies of patients. This could help to characterize neuroendocrine carcinoma of the cervix, a rare variant of cervical cancer, or help to identify cases of CC with a poor prognosis.

Although pathologists still play an important role in the development of diagnostic and therapeutic approaches, the analysis of molecular profiles at different stages of cervical cancer has enabled the development of modern diagnostic strategies, even from patient serum, which helps in establish non-invasive techniques. Currently, several molecular targets have been identified, some of which have diagnostic value, while others have prognostic and predictive value. Among these, the list of molecules is extensive, including Ki-67, cyclin D1, p53, p63, BCL-2, BCL-XL, BAX, E-cadherin, P-cadherin, CD44, ADAM9, MT1-MMP, TIMP-1, TIMP-2, MT1-MMP, MMP-2, MMP-1, MMP-9, MMP-14, proMMP-14 furin, gelatinase, TIMP-1 and TIMP-2, Nanog, nucleostemin (NS), musashi1 (Msi1), SOX2, KLF4, CD133, Cd44, ALDH1, CD49f, ABCG2, BMI1, PIWIL2, LGR5, OCT4, CD117, and others [15]. There are also several miRNAs that can be used as diagnostic, prognostic, and therapeutic biomarkers of cervical cancer [25]. Currently, the differential expression of these molecules in precancerous lesions and CC can be determined by different molecular techniques using different biological samples [26]. However, molecules that have been tested on cell lines must first be validated on a significant number of patient samples.

The possible causes of the nuclear REST expression loss in precancerous lesions and CC are not the subject of this study; however, based on the evidence reported, we suggest two alternatives that merit future study: (1) proteasomal degradation occurs after overexpression of the oncoprotein E6 in lesions with high-risk human papillomavirus, similar to the p53 elimination during cervical carcinogenesis [27]. Furthermore, proteasomal REST degradation has been documented in embryonic neurogenesis and rodent cells transformed with the adenovirus model. In these reports, REST degradation was elicited by two distinct degrons in the C-terminal, which were recognized by the E3 ubiquitin ligase SCF β-TrCP, promoting REST proteolysis [28]. (2) The production of truncated splicing isoforms lacking the nuclear localization signal (NLS) [5] results in cytoplasmic retention. Either way, the downregulation of REST could have an important effect on the transcription of its target genes.

Since REST has hundreds of molecular targets, the effect of REST downregulation on the expression of its molecular targets, observed in HeLa cells, strongly suggests that the decrease in REST, previously observed by our working group in cytology samples from a large number of patients, may serve as a platform for searching proteins and peptides that are altered in different stages of cervical cancer. The effect of REST silencing on the transcription of its target genes (Figure 2) suggests dual functions for REST (i.e., repressor or enhancer) [29] and an extensive potential role in the transcriptional dysregulation of neuronal genes in cervical cancer development, which could promote a much more aggressive phenotype during cancer progression. Technically, some of the genes analyzed in this study could be incorporated into a more extensive study to design a panel of biomarkers for the diagnosis and prognosis of cervical cancer and other REST-deficient cancers. However, because the HeLa cell line was used as a model in this study, it only allows us to have a preliminary overview of some of the genes that are proposed as biomarkers. Furthermore, its validation requires a much more robust analysis using a statistically relevant number of patient samples.

## Figures and Tables

**Figure 1 medicina-59-00537-f001:**
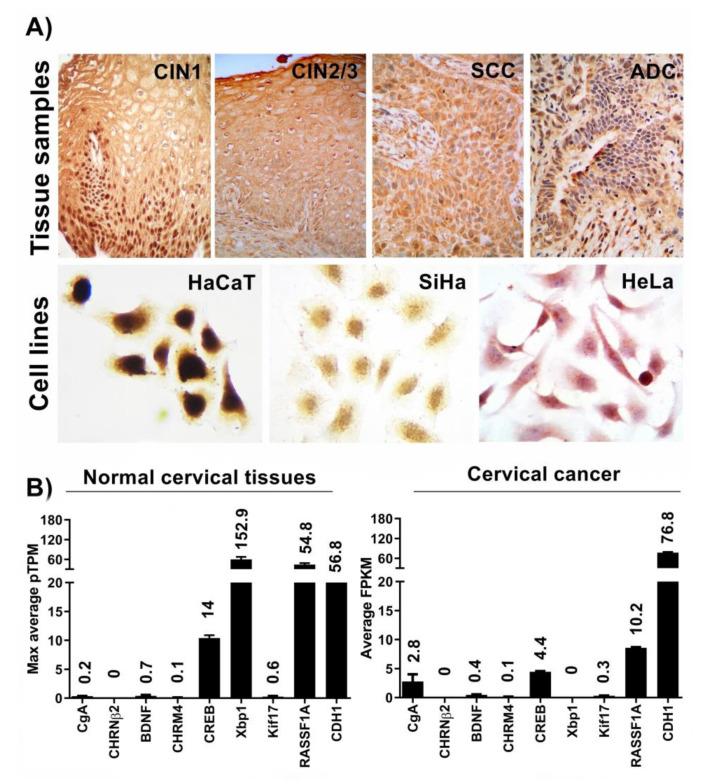
Expression of REST (RE1-Silencing Transcription factor) in tissues and cell lines. (**A**) REST immunostaining in tissue samples of premalignant lesions, cervical cancer, and cell lines. CIN: cervical intraepithelial neoplasia; SCC: squamous cell carcinoma; and ADC: adenocarcinoma. (**B**) Predicted mRNA levels of neuronal genes in normal cervical tissue (left graphic) and cervical cancer samples (right graphic), generated by the Genotype-Tissue Expression (GTEx) project from human tissues; TPM: transcripts per million; FPKM: fragments per kilobase of exon per million reads. CgA: chromogranin A; CHRNβ2: cholinergic receptor nicotinic beta 2 subunit; BDNF: brain-derived neurotrophic factor; CHRM4: cholinergic receptor muscarinic 4; CREB: CAMP responsive element binding protein 1; Xbp1: X-box binding protein 1; Kif17: kinesin-like protein; CDH1: cadherin 1; and RASSF1: Ras association domain family 1.

**Figure 2 medicina-59-00537-f002:**
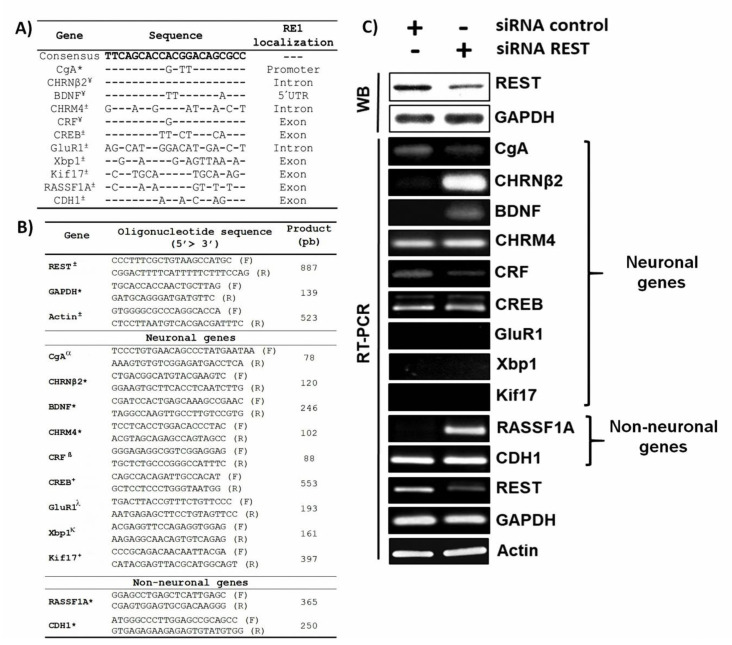
Localization of RE1 (Restrictive element 1) sequences and effect of REST silencing on its target genes in HeLa cells. (**A**) RE1 sequences in the monitored genes. * and ¥ correspond to RE1 sequences previously reported by Kashiwagi et al. (2012) [18] and Schoenherr et al. (1996) [19], respectively, while ± corresponds to RE1 sequences reported in this study. (**B**) List of oligonucleotides used in this study [20,21,22,23,24]. F: Forward; R: Reverse. * Oligonucleotides used in this study; Oligonucleotides reported by: ^± ^Ortuño-Pineda et al., [3]; ^α^ Chteinberg et al., 2018 [20]; ^β^ Janitzky et al., 2014 [21]; ^+^ Bo et al., 2014 [22]; ^λ^ Challenor et al., 2015 [23]; ^κ^ Wang et al., 2015 [24] (**C**) REST silencing and RT–PCR of the REST targets. The upper panel labeled WB corresponds to Western blot for REST, and the lower panel corresponds to RT–PCR for REST target genes. CRF: corticotropin-releasing factor; GluR1: ionotropic glutamate receptor.

## Data Availability

Not applicable.

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
