# Peer review of "REST/NRSF Silencing Modifies Neuronal Gene Expression in siRNA-Treated HeLa Cells: A Preliminary Exploration in the Search for Neuronal Biomarkers of Cervical Cancer"

_medicina, 2023, doi:10.3390/medicina59030537_

Round 1
Reviewer 1 Report
A communication by C. Ortuño-Pineda and coworkers reports on the role of RE1-silencing transcription factor (REST) in cervical cancer. The topic is important because of especially frequent occurrence of cervical cancer in women worldwide and the need to develop new early diagnostic procedures to recognize the malignancy. The paper is well-written, the results have been presented in a concise way, conclusions are supported by the available evidence. I would recommend acceptance after revision, specifically when the points specified below are addressed.
Specific comments
1. There is no phenotypic characterization for the tissues being studied. The authors say that the tissues with reduced REST expression acquire a neuroendocrine phenotype according to the literature. What is known for the tissues they are examining and in particular for the samples of cervical cancer from the literature on the subject? The authors themselves did not determine the phenotype of the samples.
2. The authors did not specify how many samples are statistically reliable in order to draw conclusions about the suitability of using the markers mentioned in the article as diagnostic markers. There are no error bars in determining the values in Fig. 1, the authors gave only the averages. It was not specified how much the values differed between specific samples.
3. Please increase the font in Fig. 2B as the sequences are not readable.
4. Authors should be more specific about how they present diagnostics based on their data. What specific markers can be used and how? What needs to be investigated, the level of expression, the level of proteins? Are there such assays in the clinical practice specifically for these markers, or methods need to be developed?
Author Response
Point 1: There is no phenotypic characterization for the tissues being studied. The authors say that the tissues with reduced REST expression acquire a neuroendocrine phenotype according to the literature. What is known for the tissues they are examining and in particular for the samples of
cervical cancer from the literature on the subject? The authors themselves did not determine the phenotype of the samples.
Response 1:
Regarding the comment on the phenotypic characteristics of the samples, we added the following sentences into the text (“Material and methods” and “Results” sections, respectively): “Histopathological diagnosis was performed with the H&E stain and analyzed by an experimented pathologist based on the cervical intraepithelial neoplasia classification”
“The phenotypic characterization of tissues was performed by H&E staining by a pathologistand classifies as follow: in CIN 1 cells with binucleation, perinuclear halo and caryomegaly are observed, and in CIN 2/3 and CC cells have caryomegaly, dyskaryosis, hyperchromatic nuclei and granular chromatin.”
Regarding the question “What is known for the tissues they are examining and in particular for the samples of cervical cancer from the literature on the subject?”, we provide the following argumentation for the reviewer:
A clear example of the role of REST in the overexpression of neuronal peptides and proteins is the small cell lung cancer (SCLC), where the reduced expression of REST has been proposed as a cause of the neuroendocrine phenotype overexpressing mainly chromogranin, vasopressin,
synaptophysin, as markers of neuronal phenotype (Coulson et al., 2000). In the case of cervical cancer, the expression of REST has not been widely explored, however, our group recently reported the loss of REST expression in high-grade premalignant lesions and cervical cancer,
but no neuronal target of REST was explored (Cortés-Sarabia et al., 2021). It has been reported that positive immunohistochemical staining for neuroendocrine markers such as synaptophysin, chromogranin, CD56, and neuron-specific enolase is diagnostic for
neuroendocrine carcinoma of the cervix (NECC), a rare of cervical cancer, but not for squamous cell carcinoma or adenocarcinoma (Clemens et al., 2018). In this study, we do not propose that REST-deficient samples present a neuroendocrine phenotype, but rather that alteration of REST
may change the expression of some of its target genes, which could have potential use in diagnosis.
-Coulson, J. M., Edgson, J. L., Woll, P. J. & Quinn, J. P. A splice variant of the neuron-restrictive silencer factor repressor is expressed in small cell lung cancer: a potential role in derepression of neuroendocrine genes and a useful clinical marker. Cancer Res. 2000;60:1840–1844.
-Cortés-Sarabia, K, Alarcón-Romero, L.C., Flores-Alfaro, E., Illades-Aguiar, B., VencesVelázquez, A., Mendoza-Catalán, M.A., Navarro-Tito, N., Valdés, J., Moreno-Godínez, M.E. & Ortuño-Pineda, C. Significant decrease of a master regulator of genes (REST/NRSF) in highgrade squamous intraepithelial lesion and cervical cancer. Biomedical Journal. 2021;44:S171-
S178.
-Clemens B. Tempfer, Iris Tischoff, Askin Dogan, Ziad Hilal, Beate Schultheis, Peter Kern and Günther A. Rezniczek. Neuroendocrine carcinoma of the cervix: a systematic review of the literatura. BMC Cancer (2018) 18:530
Point 2: The authors did not specify how many samples are statistically reliable in order to draw conclusions about the suitability of using the markers mentioned in the article as diagnostic markers.
There are no error bars in determining the values in Fig. 1, the authors gave only the averages. It was not specified how much the values differed between specific samples.
Response 2:
Regarding the comment, “The authors did not specify how many samples are statistically reliable in order to draw conclusions about the suitability of using the markers mentioned in the article as diagnostic markers,” we added the following sentence into the text (at the end of the “Results”
section):“Because we use the HeLa cell line as a model, the reliability of using such markers in diagnosis
will depend on testing them extensively in a statistically appropriate number of patient samples.”
Regarding the comment, “There are no error bars in determining the values in Fig. 1, the authors gave only the averages. It was not specified how much the values differed between specific
samples”: We added error bars in figure 1 in the manuscript. In addition, we provide the following argumentation for the reviewer: although the units of measurement for cancer samples (TPM:
Transcripts Per Kilobase Million) are different from those for normal cervical samples (FPKM: Fragments Per Kilobase Million), TPM is very similar to FPKM, the only difference is the order of operations for normalization. Due to possible small discrepancies in units of measurement,
we did not make numerical comparisons. However, the analysis gives us a consistent idea of the altered expression of neural genes in REST-deficient cancer samples that could have potential use in diagnosis. In particular, we observed a prominent change in expression for CgA, CREB, Xbp1, and others such as RASSF1A, and CDH1.” Point 3: Please increase the font in Fig. 2B as the sequences are not readable.
Response 3:
We increased the font size in figure 2B.
Point 4: Authors should be more specific about how they present diagnostics based on their data. What specific markers can be used and how? What needs to be investigated, the level of expression, the level of proteins? Are there such assays in the clinical practice specifically for these markers, or methods need to be developed?
Response 4:
Regarding the comment, “Authors should be more specific about how they present diagnostics based on their data. What specific markers can be used and how?”
We added the next sentence into the text (in the “Discussion” section): “Although the analysis shown in figure 1B corresponds mainly to data obtained from squamous cell carcinoma samples and the results in Figure 2C correspond to the HeLa cell line, which derives from an adenocarcinoma; in both cases we observed alteration in the expression of some neuronal genes. Remarkably, it was observed an increase in the expression of the CHRNβ2 neuronal
gene, which has been widely reported as a transcriptional target of REST and could be included in the panel of cervical cancer prognostic biomarkers.”
Regarding the comment, “What needs to be investigated, the level of expression, the level of proteins? Are there such assays in the clinical practice specifically for these markers, or methods
need to be developed?”
We added the next sentence into the text (in the “Discussion” section): “In the first instance, some of the neuronal genes with altered expression could be included in the panel of biomarkers whose protein levels could be observed by Immunohistochemistry in formalinfixed, paraffin-embedded block of tissue using monoclonal antibodies (commercially available) against each protein. Additionally, an RT-PCR could be standardized to monitor the
RNA levels of these genes from cervical cytologies of patients. This could help to characterize neuroendocrine carcinoma of the cervix, a rare variant of cervical cancer, or would help to identify cases or CC with poor prognosis.”
Reviewer 2 Report
A very interesting argoument well treated by Authors and well write.
Author Response
This reviewer did not make any specific comments.
Reviewer 3 Report
Cortes-Sarabia and Co-workers presented a manuscript entitled “REST/NRSF silencing modifies neuronal gene expression in siRNA-treated HeLa cells: a preliminary exploration in the search for neuronal biomarkers of cervical cancer” in which the authors set out to investigate which neural genes have an altered expression following REST/NRSF silencing in Hela cells. Deregulation of the expression of these genes could serve as a molecular target for the diagnosis of cervical cancer, in which reduced expression of REST has been already demonstrated by the authors. Even if it deals with a topic of relevant interest, that is the identification of biomarkers for the diagnosis of cervical cancer, it does not present sufficient novelty for publication. Indeed the experiment showed in Figure 1a has been already published by authors (Cortés-Sarabia K, Alarcón-Romero LDC, Flores-Alfaro E, Illades-Aguiar B, Vences-Velázquez A, Mendoza-Catalán MÁ, Navarro-Tito N, Valdés J, Moreno-Godínez ME, Ortuño-Pineda C. Significant decrease of a master regulator of genes (REST/NRSF) in high-grade squamous intraepithelial lesion and cervical cancer. Biomed J. 2021 doi: 10.1016/j.bj.2020.08.012. Epub 2020 Aug 27. PMID: 35491677; PMCID: PMC9068566). The second experiment performed by authors consists in the evaluation of mRNA levels of target genes in Hela cells silenced for REST. In my opinion this experiment does not provide sufficient novelty as gene expression was only evaluated in an immortalised cell line and provides contrasting results to Figure 2b, which shows the mRNA levels of the target genes in cervical cancer tissues respect to normal cervical tissues obtained by Human Protein Atlas. Moreover, the y-axis of graph in figure 2b present a different unite of measurement, making it difficult to compare the two conditions.
Author Response
Point 1: Even if it deals with a topic of relevant interest, that is the identification of biomarkers for the diagnosis of cervical cancer, it does not present sufficient novelty for publication. Indeed the experiment showed in Figure 1a has been already published by authors (Cortés-Sarabia K, AlarcónRomero LDC, Flores-Alfaro E, Illades-Aguiar B, Vences-Velázquez A, Mendoza-Catalán MÁ, NavarroTito N, Valdés J, Moreno-Godínez ME, Ortuño-Pineda C. Significant decrease of a master regulator of genes (REST/NRSF) in high-grade squamous intraepithelial lesion and cervical cancer. Biomed J.
2021 doi: 10.1016/j.bj.2020.08.012. Epub 2020 Aug 27. PMID: 35491677; PMCID: PMC9068566).
Response 1: Although we reported for the first time in 2021 a loss of REST expression in premalignant lesions of the cervix and in cervical cancer in a large sample of patients, in the present work we used samples from the same collection but from different patients, and we clearly mention
that we have already published a more extensive work. Actually, figure 1a only serves as a framework to show the decrease in REST expression in such samples, but does not represent the central figure of this investigation.
Point 2: The second experiment performed by authors consists in the evaluation of mRNA levels of target genes in Hela cells silenced for REST. In my opinion this experiment does not provide sufficient novelty as gene expression was only evaluated in an immortalised cell line and provides contrasting
results to Figure 2b, which shows the mRNA levels of the target genes in cervical cancer tissues respect to normal cervical tissues obtained by Human Protein Atlas.
Response 2: We believe that the result is novel because the neuronal genes targeted by REST have not been explored in any type of cervical cancer (Squamous Cell Carcinoma, endocervical
Adenocarcinoma or Neuroendocrine Carcinoma of the Cervix). Although some neuronal markers have been explored in Neuroendocrine Carcinoma of the Cervix, their relationship with the loss of REST is unknown. Furthermore, the impact of decreased REST on the expression of its transcriptional targets in premalignant lesions of the uterine cervix and in cervical cancer has not
been studied. If we consider that REST regulates hundreds of human genes, it could be a very powerful diagnostic tool. This first work using HeLa cells constitutes preliminary evidence of this hypothesis and we will continue working to elucidate its true value in the diagnosis of this type of cancer.
In future works we may include a more comprehensive model using all cervical cancer cell lines and monitoring REST transcriptional targets in patient samples.
Point 3: Moreover, the y-axis of graph in figure 2b present a different unite of measurement, making it difficult to compare the two conditions.
Response 3: Regarding to this comment, we provide the following argumentation: We reviewed the data derived from the graph in figure 2b. Although the units of measurement for cancer samples (TPM: Transcripts Per Kilobase Million) are different from those for normal cervical
samples (FPKM: Fragments Per Kilobase Million), TPM is very similar to FPKM, the only difference is the order of operations for normalization. If the results cannot be compared as precisely, they give
us a consistent idea of the altered expression of neural genes in REST-deficient cancer samples that could have potential use in diagnosis.
Round 2
Reviewer 3 Report
I thank the authors for their reply. However, I believe that the data presented in this paper are extremely preliminary, and need to be confirmed to elucidate their true value in the diagnosis. For these reason I believe that the data presented must be strong and consistent, which is not guaranteed by a one-time RT-PCR. Therefore, I recommend that the authors confirm the results by real-time PCR accompanied by an appropriate statistical analysis indicating the altered expression of the genes considered following REST factor silencing.
Author Response
Point 1.“English language and style are fine/minor spell check required”
Response 1 We attended to this point in detail through the MDPI services (English editing ID: English61891). Below are shown in yellow all the corrected errors, which were also marked in yellow in the corrected manuscript.
ABSTRACT:
“has been barely estudied in cervical” changed to “has barely been estudied in
cervical”
“We previously reported reduced expression” changed to “We previously reported the reduced expression”
“in cancerous cells with reduced expression of REST” changed to “in cancerous
cells with reduced expression levels of REST”
“and their derivative cell lines SiHa” changed to “and the derivative cell lines SiHa”
“The results showed REST diminution” changed to “The results show REST
diminution”
“the role of REST loss in transcriptional regulation” changed to “the role of REST
loss in the transcriptional regulation”
INTRODUCTION
“In different neural contexts, the REST diminution involves” changed to “In different
neural contexts, REST diminution involves”
“the absence of REST allows expression of” changed to “the absence of REST
allows the expression of”
“some of them under the transcriptional control” changed to “some of which are
under the transcriptional control”
“the impact of the REST diminution” changed to “the impact of REST diminution”
MATERIAL AND METHODS
“the tissue samples were collected by a pathologist from female patients from
southern Mexico” changed to “the tissue samples were collected from female
patients from southern Mexico by a pathologist”
“Ethics Committee of the UAGro and IECan” changed to “The Ethics Committee
of the UAGro and IECan”
“was performed with the H&E stain and analyzed by an experimented pathologist”
changed to “was performed with H&E staining and analyzed by an experienced
pathologist”
“Paraffin tissues sections of 3 µm thick” changed to “Paraffin tissue sections of 3µm in thickness”
“In both cases ICC and IHC, we performed the streptavidin-biotin-peroxidase
method” changed to “In both cases, ICC and IHC, we used the streptavidin-biotinperoxidase method”
“Gene sequences analyzed in this study” changed to “The Gene sequences
analyzed in this study”
“reports performed by Kashiwagi et al. 2012 [18], Schoenherr et al. 1996 [19]”
changed to “reports performed by Kashiwagi et al. 2012 [18] and Schoenherr et al.1996 [19]”
“western blot and RT-PCR analysis” changed to “Western blot and RT-PCR
analysis.”
“RT-PCR was performed using oligonucleotides” changed to “RT-PCR was
performed using the oligonucleotides”
“Ethidium bromide” changed to “ethidium bromide”
RESULTS
“These findings suggested a potential effect of REST diminution” chancged to
“These findings suggest a potential effect of REST diminution”
“To analyze this, we first evidenced expression profiles of REST” changed to “To analyze this association, we first generated expression profiles of REST”
“as well as in HeLa and SiHa cell lines” changed to “as well as in the HeLa and
SiHa cell lines”
“The phenotypic characterization of tissues was performed by H&E staining by a pathologist and classifies as follow” changed to “The phenotypic characterization of tissues was performed using H&E staining by a pathologist and can be classified as follows”
“in CIN 1 cells with binucleation, perinuclear halo and caryomegaly are observed, and in CIN 2/3 and CC cells have caryomegaly, dyskaryosis, hyperchromatic nuclei and granular chromatin” changed to “in CIN 1 cells with binucleation, perinuclear halo, and karyomegaly are observed, and in CIN 2/3 and CC cells karyomegaly, dyskaryosis, hyperchromatic nuclei, and granular chromatin are observed”
“Results confirmed the evident nuclear REST diminution” changed to “The results confirm the evident nuclear REST diminution”
“as it was reported by other groups and our in silico analysis” changed to “as
reported by other groups and our in silico analysis”
“Because we use the HeLa cell line” changed to “Because we used the HeLa cell line”
DISCUSSION
“Some examples of cancers with reduced expression of REST include lung”
changed to “Some examples of cancers with reduced expression levels of REST
include lung”
“CIN 2/3, SCC and ADC samples, as well as cancerous cells showed diminution in the REST expression (Figure 1)” changed to “CIN 2/3, SCC, and ADC samples,
as well as cancerous cells, showed diminution in their REST expression (Figure 1)”
“Also, further REST diminution in HeLa cells” changed to “Additionally, further
REST diminution in HeLa cells”
“suggesting a potential role for REST on the alteration of many neural genes in
cancer, with application in diagnosis” changed to “suggesting a potential role of
REST in the alteration of many neural genes in cancer, with potential application in diagnosis”
“correspond to the HeLa cell line, which derives from an adenocarcinoma;”
changed to “correspond to the HeLa cell line, which derives from an
adenocarcinoma,”
“Remarkably, it was observed an increase in the expression of the CHRNβ2
neuronal gene in HeLa cells,” changed to “Remarkably, an increase in the
expression of the CHRNβ2 neuronal gene in HeLa cells was observed,”
“could be observed by Immunohistochemistry in formalin-fixed, paraffin-embedded block of tissue using monoclonal antibodies (commercially available) against each protein” changed to “could be observed by Immunohistochemistry in a formalin fixed, paraffin-embedded block of tissue using (commercially available) monoclonal
antibodies against each protein.”
“from cervical cytologies of patients” changed to “from the cervical cytologies of patients”
“or would help to identify cases or CC with poor prognosis” changed to “or would help to identify cases of CC with poor prognosis”
“which allow establishing non-invasive techniques” changed to “which allows the establisment of non-invasive techniques”
“promoting the REST proteolysis” changed to “promoting REST proteolysis”
“Production of truncated splicing isoforms lacking the nuclear localization signal(NLS) [5], resulting in cytoplasmic retention” changed to “The production of truncated splicing isoforms lacking the nuclear localization signal (NLS) [5] results in cytoplasmic retention”
“Either way, downregulation of REST” changed to “Either way, the downregulation of REST”
“here observed in HeLa cells” changed to “observed here in HeLa cells”
“dual functions for REST as repressor or enhancer” changed to “dual functions for REST as a repressor or enhancer”
“potential role in transcriptional dysregulation” changed to “potential role in the transcriptional dysregulation”
The following sentence was added at the end of the discussion: However,
because the HeLa cell line used as a model in this study only allows us to have a
preliminary overview of some of the genes that are proposed as biomarkers, its
validation requires a much more robust analysis using a statistically relevant
number of patient samples.
Point 2. “I thank the authors for their reply. However, I believe that the data presented in this paper are extremely preliminary, and need to be confirmed to elucidate their true value
in the diagnosis. For these reason I believe that the data presented must be strong and consistent, which is not guaranteed by a one-time RT-PCR. Therefore, I recommend that the authors confirm the results by real-time PCR accompanied by an appropriate statistical
analysis indicating the altered expression of the genes considered following REST factor silencing”.
Response 2
Certainly, the siRNA-treated HeLa cells used as a model in this study only allow us to present a preliminary overview of some potential biomarkers, but the validation requires a much more robust analysis. Unfortunately, it is not possible for us to carry
out further experiments for now. However, we think that the evident change in the expression levels of some of the REST transcriptional targets (CHRNβ2, BDNF, CRF, and even RASSF1a) allow us to propose them as potential biomarkers.
To avoid discrepancies between the results of cell lines and patient samples, the
results must be validated on a statistically relevant number of cervical smears and biopsies. In the clinic, the most used techniques are immunocytochemistry and immunohistochemistry, since they allow the determination of the decrease or overexpression of proteins and also the phenotypic characterization of the tumor.
These techniques can be complemented with real-time PCR for a better
characterization. Because this work is part of a project that we are concluding (doi: 10.1093/jb/mvz046 and doi: 10.1016/j.bj.2020.08.012), it is not possible for us to address a more robust analysis using other experimental approaches, or other cancer cell lines and a significant number of patient samples for now.